# Biocomposites of Epoxidized Natural Rubber Modified with Natural Substances

**DOI:** 10.3390/molecules27227877

**Published:** 2022-11-15

**Authors:** Konrad Stefaniak, Anna Masek, Aleksandra Jastrzębska

**Affiliations:** 1Institute of Polymer and Dye Technology, Faculty of Chemistry, Lodz University of Technology, Stefanowskiego 16, 90-537 Lodz, Poland; 2Institute of Materials Science and Engineering, Faculty of Mechanical Engineering, Lodz University of Technology, Stefanowskiego 1/15, 90-537 Lodz, Poland

**Keywords:** ENR, natural substances, catechin hydrate, flavone, solar aging

## Abstract

This research aimed to show the possible impact of natural antioxidants on epoxidized natural rubber (ENR) and poly(lactic acid) (PLA) green composites. Thus, the ENR/PLA blends were prepared with the addition of three selected phytochemicals (catechin hydrate, eugenol and flavone). Obtained materials were submitted for solar aging. The analysis of the samples’ features revealed that catechin hydrate is a natural substance that may delay the degradation of ENR/PLA blends under the abovementioned conditions. The blend loaded with catechin hydrate presented stable color parameters (dE < 3 a.u.), the highest aging coefficient (K = 0.38 a.u.) and the lowest carbonyl index based on FT-IR data (CI = 1.56) from among all specimens. What is more, this specimen prolonged the oxidation induction time in comparison with the reference samples. Gathered data prove the efficiency of catechin hydrate as an anti-aging additive. Additionally, it was found that a specimen loaded with flavone changed its color parameters significantly after solar aging (dE = 14.83 a.u.) so that it would be used as an aging indicator. Eventually, presented eco-friendly ENR-based compositions may be applied in polymer technology where materials presenting specific properties are desirable.

## 1. Introduction

Composite materials based on biopolymers are appreciated for their eco-friendly character. Such materials can consist of bio-based elastomers and biodegradable thermoplastics. Moreover, they can be modified by adding natural antioxidants to them to obtain materials that are more resistant to aging processes. Flavonoids, which belong to the group of phenols and polyphenols, are especially significant antioxidants as they present antiradical activity and inhibit oxidation processes. The activity of phytochemicals depends on their chemical structure, e.g., the number of hydroxyl groups in a molecule.

Epoxidized natural rubber/poly(lactic acid) (ENR/PLA) blends are reported to be one of the most interesting composite materials [1,2,3,4]. Obviously there are many other different green composite materials based on compounds such as polyvinyl alcohol (PVA) [5], polycaprolactone (PCL) [6] and gelatin [7,8], but choosing PLA is preferable. It results from the derivation of each material. Both PVA and PCL are based in the petroleum industry, which is harmful to the environment [9,10]. Abundant gelatin sources are derived from animals [8], which could be used in the food industry rather than polymer technology. However, PLA is a biodegradable and biocompatible aliphatic polyester that derives from renewable, plant-based resources, and in that it stands out from other polymers as a bioplastic [11]. This paper presents the application of natural antioxidants into thermoplastic ENR/PLA blends during processing of the materials.

Natural antioxidants can be defined as naturally occurring molecules diversely present in living organisms. They feature several biological effects—antibacterial, anti-aging, antiviral, anti-inflammatory and anticancer activities [12]. What is more, natural antioxidants react with free radicals to reduce their amount and at the same time decrease oxidative stress [13,14]. Currently, natural antioxidants, apart from being used in the food and pharmaceutical industries, have been studied as suitable and efficient stabilizers for the protection of polymers and biopolymers from oxidative degradation [15,16,17,18,19]. Natural antioxidants can be classified and divided into particular groups (Figure 1 [20]). Chemical compounds that can be used as appropriate stabilizers are mainly phenols, polyphenols [21], phenolic acid derivatives [22], vitamins [23] and carotenoids [24,25]. Considering polyphenols, it is proposed that their radical scavenging activity consists of transferring hydrogen atoms from the phenolic hydroxyl group to the radicals [12]. The number and position of phenolic hydroxyl groups are the reasons that determine the activity of a phenolic antioxidant molecule as it is agreed that the main mechanism of the radical scavenging effect of phenolic compounds is hydrogen atom transfer from the phenolic hydroxyl moiety to the reacting radical [26].

Agustin-Salazar et al. [27] introduced resveratrol (a naturally occurring stilbenoid) in PLA film and found out that as a function of UV exposure time, reductions of elongation at break (ε) and number-average molecular weight (M_n_) for PLA/resveratrol systems were less distinct compared to the reductions of ε and M_n_ for neat PLA. After 30 h of irradiation, the M_n_ of neat PLA was lowered by circa 60%, while PLA/resveratrol specimens retained approximately 75% of their original molar mass. Furthermore, after about 60 h of irradiation, neat PLA specimens were unfit to be tested in terms of mechanical properties (they were too brittle to be handled) whereas PLA/resveratrol samples retained approximately 30% of their initial ε.

Other interesting research concerned using α-tocopherol (vitamin E) as an adequate processing stabilizer against the thermo-oxidative degradation that occurs, e.g., during the processing of poly(lactic acid) for food packaging [23,28,29]. It reported that adding α-tocopherol to polylactic acid/polyethylene glycol (PLA/PEG) films resulted in a 23.7% decrease in water vapor permeability and 9.4% increase in oxygen permeability in relation to PLA/PEG without α-tocopherol. This second observation limits using PLA/α-tocopherol films as efficient food-packaging protectors. In addition, PLA/α-tocopherol films showed enhanced antibacterial and antioxidant features. Campoccia et al. [30] studied bacterial adhesion of PLA/α-tocopherol films. Results revealed that the incorporation of α-tocopherol in PLA films led to a 41% decrease of *S. aureus* adhesion in Tryptose Broth medium after 24 h in comparison to neat PLA.

Furthermore, natural antioxidants can be used as aging indicators. López-Rubio et al. [31] proposed using β-carotene (carotenoid characterized by intense orange-red color) to this end. Observed rapid color change in PLA/β-carotene films might be utilized as an important parameter to follow the degradation that appears as a result of UV-light exposure.

In this work, the influence of adding three different phenolic compounds to ENR/PLA composites was scrutinized. These substances were: eugenol, catechin hydrate and flavone. These chemical compounds were selected due to their different molecular composition in terms of the number of hydroxyl groups in the particle. Catechin hydrate and eugenol have five and one hydroxyl groups, respectively. Flavone has not any hydroxyl groups in its composition, but as it belongs to the group of flavonoids it should potentially present antioxidant activity.

Recently, these compounds have been studied regarding their use in polymer technology. Sabaa et al. [32] proved that eugenol can be used as an efficient thermal stabilizer for rigid polyvinyl chloride (PVC). Adding eugenol to PVC resulted in a threefold increase in thermal stability compared to PVC stabilized by tin mercaptide (reference stabilizer). Eugenol’s high stabilizing efficiency results from its relatively high thermal-stability value. Catechin hydrate nanoparticles together with chitosan-coated-poly(lactic-co-glycolic acid)-nanoparticles as a drug-delivery system were used in order to improve brain and lung bioavailability for catechin hydrate regarding lung cancer and epilepsy treatments [33,34]. Other studies were presented by Abral et al. [35]. Ultrasonicated cellulose-based biocomposites with an extract from Uncaria gambir (G) leaves, which contain polyphenols, predominantly catechin, were obtained. The presented materials have potential for packaging applications. The results showed that adding 0.05 g of G per gram of bacterial cellulose (BC) powder increased the value of a specimen’s tensile strength from 72.9 MPa to 105.6 MPa. Moreover, adding 0.2 g of G per 1 g of BC powder enhanced the antimicrobial activity of the material (zone of inhibition against *Escherichia coli* increased from 0 mm to 6.8 mm).

The ENR-based materials described in this paper also included in their composition hydroxypropyl methylcellulose (HPMC), which acted as an immobilizer of added antioxidants. Apart from cellulose derivatives, there are several other substances used in polymer technology as immobilizers. Sahiner et al. [36] immobilized tannic acid (TA) on silica nanoparticles. Derived nanocomposites presented good antioxidant activity. Kuzema et al. [37] adsorbed vitamins C and E on nanosilica compounds. Vitamins immobilized on a silica surface presented their stabilization. Prolonged desorption of vitamins was observed. Other commonly used immobilizers are carbon nanotubes and graphene oxide [38].

The aim of this study was to obtain ENR/PLA blends with natural antioxidants—eugenol, catechin hydrate, flavone—and an immobilizer of natural additives in the form of HPMC and analyze the influence of these additives on materials features compared to samples without these antioxidants.

The manufactured specimens were subjected to a solar-aging process and the effects of selected additives on the several properties of ENR/PLA blends have been studied. This research contributes to extending the knowledge about incorporating natural antioxidants into polymer matrices and broadening this branch of science. Selected natural antioxidants presented in this paper had not previously been used as additives for ENR/PLA blends. Compositions of created materials and studying the impact of additives on retarding the degradation processes of polymer matrix under conditions of solar aging are scientific novelties of this research. What is more, ecological HPMC used as an immobilizer of antioxidants has the potential to improve its dispersal in a biocomposite and, as a result, may enhance its antioxidative activity during the aging process. Considering the chemical structures of eugenol, catechin hydrate and flavone, these natural substances may act as pro-ecological additives for polymeric materials able to improve their physico–mechanical properties.

## 2. Results and Discussion

### 2.1. Change of Color Measurement

Natural phenolic compounds can indicate a composite’s color change after aging. Hence, determining changes that occurred regarding visual features of studied specimens can be classed as very assistant data. According to the color change index (dE) presented in Figure 2A, composites with the addition of eugenol and flavone and the ENR–PLA composite showed the highest color change indices after solar aging, where dE equalled respectively (11.00 ± 1.69) arbitrary unit (a.u.), (14.83 ± 0.61) a.u., (12.48 ± 1.06) a.u. This means that an inexperienced observer can see the difference and the above composites can be used as aging indicators. Oppositely can be described composites with catechin hydrate and the ENR–CEL composite—their dE was quite low and averaged between 2.00 a.u. and 3.00 a.u. This means that these composites are the most stable regarding the change of color. Observed changes of color were caused by the implemented solar radiation (radical reactions were initiated). Furthermore, aged composites commonly present changes in their chemical structure—for this reason the FT-IR spectrum is examined in what follows. Irreversible modifications within the polymer chain could also occur—crosslinking, scission and oxidation.

Moreover, the activity of an antioxidant is contingent on its chemical structure. The presence of hydroxyl moieties inhibits the aging process. Catechin hydrate has five hydroxyl groups in its structure—more than flavone and eugenol. Therefore, the ENR–CAT composite presented a lower change of color than composites including two other natural antioxidants. It can be assumed that the single -OH group present in eugenol structure and the distinguishing characteristic of flavone compound (carbonyl group and conjugated double bonds systems) do not present sufficient antioxidative activity in order to stabilize the materials during accelerated aging.

Figure 2B presents data relative to the whiteness index. It can be noted that the specimen with catechin hydrate was the darkest and the PLA–CEL composite was the lightest. This is the effect of additives’ original color. What is more, for all specimens with the exception of PLA–FLA, solar aging led to the increase in the whiteness index, which reveals that specimens became lighter after aging.

Figure 2C enables analysis of the chroma of specimens. It was characteristic for ENR–FLA and ENR–CAT composites that, as a result of solar aging, their chroma values increased—the color of specimens became more brilliant and saturated. Composites with the addition of eugenol and the ENR–PLA specimen behaved inversely; the decrease in saturation was observed. It can be noted that chroma varied in inverse proportion to the whiteness index parameter when the impact of solar aging was studied. Specimens with eugenol and ENR–PLA became lighter after aging, but their color was less saturated concurrently. This can result from a character of solar radiation. A similar phenomenon was spotted for the ENR/PLA composite with the addition of δ-tocopherol [39].

Hue angle parameters are presented in Figure 2D. It can be observed that apart from ENR–CAT specimens, all examined specimens were characterized with hue angle values between 80° and 100°. This means that their colors were close to yellow and orange shadings. The ENR–CEL specimen was the most yellowish. The ENR–CAT composite differed considerably. Its hue angle value suggested that its color was close to red shading. Regarding the effect of solar aging, the hue angle increased for all specimens or remained nearly the same.

Conducted measurements concerning optical properties of studied materials assisted in measuring the color stability during the aging process. The effectivity of stabilization induced by adding natural additives to the polymer matrix could be evaluated. As a result of the aging process, the oxidation of materials occurs, which leads to samples yellowing or browning. It is the evidence of carbonyl groups being created. Hence, it can be noted that ENR–EUG, ENR–PLA, and ENR–FLA samples were the least stable in terms of color change. These samples presented the most significant color change towards a yellow taint after solar aging, which can indicate occurring degradation. This deterioration of the polymer surface implies intense interaction of the specimen with solar radiation. Compounds present in the specimen decomposed in some degree and as a result the color became brighter—other wavelengths of the light were absorbed. On the other hand, ENR–CEL and ENR–CAT samples did not present a visible change in color. Individual parameters describing their color stayed almost unchanged in comparison with other samples. It confirms their color stability and limitations in the matter of carbonyl moieties creation. A hydrogen atom from a hydroxyl group (which is present, e.g., in catechin hydrate) can be transferred to the free radical, which results in inhibiting the oxidation mechanism of the polymer. Hence, the change of color is little.

### 2.2. Mechanical Properties

The static mechanical tests were carried out to characterize the tensile strength (TS) and elongation at break (Eb) of the analyzed ENR-based compositions (Figure 3). The aim of these tests was to study the impact of solar aging on specimens’ properties. Regardless of the findings, the obtained results give valuable information concerning the stabilizing activity of selected antioxidants. Additionally, three different aging coefficients were calculated (Table 1 and Figure 4). According to Figure 3, a decrease in TS and Eb values can be observed for all specimens after aging. It is a result of solar radiation absorbed by polymers during the process of aging.

The most significant decrease in TS value can be perceived for the ENR–PLA sample. It fell from (4.28 ± 0.52) MPa to (1.42 ± 0.05) MPa. Parenthetically, this material presented the highest value of TS before aging out of all studied materials. At the other end of the spectrum, the lowest values of TS were presented by the ENR–CEL sample. Its TS was (1.87 ± 0.06) MPa before aging and (0.98 ± 0.12) MPa after aging. It should be highlighted that ENR–FLA showed the lowest decrease of TS after aging in comparison with other blends (fall from (2.15 ± 0.23) MPa to (2.01 ± 0.05) MPa). It means that using flavone as an additive caused higher aging stability of the material. It can be assumed that this phenomenon is a result of sufficient cross-linking density occurring between the flavone and the polymer matrix initiated with the solar-aging process. Considering the concurrent drop in elongation at the break of the ENR–FLA sample, it can be assumed that the composite became stiffer and lost its flexibility to some extent. It can be explained by the chemical interaction that appeared between flavone particles and the polymer matrix. Similar findings were related to other antioxidant substances such as quercetin or green tea extract [39,40,41]. What is more, the anti-aging potential of flavone is much unsupported by its chemical structure as it has not any hydroxyl groups responsible for antioxidant activity.

Regarding the results of tests considering elongation at break, again all Eb values of individual samples decreased after aging. ENR–PLA showed the finest findings out of all studied materials. Its Eb was (683 ± 37)% before aging and (196 ± 19)% after aging. The biggest decrease in Eb values was presented by the ENR–FLA sample. Its Eb after aging was 5.4 times lower than before aging (fell from (368 ± 62)% to (68 ± 6)%).

Three different aging coefficients were calculated. K refers to TS and Eb, K1 refers only to TS and K2 refers only to Eb. In general, the analysis of K values revealed that prepared compositions do not present high aging stability as their K aging coefficients are much lower than 1.00 a.u. What is more, ENR/PLA had the lowest K (0.10 a.u.) and it was the only specimen that did not contain cellulose fibers. It suggests that adding cellulose fibers to ENR-based blends has an impact on mechanical properties of material when it is submitted to solar aging. The highest K value (0.38 a.u.) was presented by the ENR/CAT composite. It was caused by a good antioxidant activity of catechin hydrate as it has five hydroxyl groups in its structure. Potential crosslinks created under the aging process in ENR matrix by catechin hydrate might be the reason for mechanical properties’ improvement as well.

To sum up the discussion of mechanical properties, it should be noted that TS, Eb and K results correlate with each other. The most thought-provoking is the performance of the ENR–FLA specimen concerning its unique TS value after aging but very low Eb after aging as well.

In terms of aging impact on mechanical properties, the ENR–CAT sample seems to be the most stable. It presented the highest values of K and K2 coefficients from among all specimens. It could be a result of catechin’s antioxidative activity and its potential crosslinking with the polymer matrix. These data correlate with the change of color measurement, which also suggested suitable resistance of the ENR–CAT sample to solar aging.

Additionally, it should be highlighted that the ENR–EUG sample has potential to be used in the packaging industry despite a very intensive smell of pure eugenol. This smell is significantly reduced as a result of material processing. Eventually, it was verified via organoleptic analysis that aged samples had lost their smell and there is not any contraindication to usage in the packaging industry.

### 2.3. Thermogravimetric Analysis (TGA)

Performed thermogravimetric analysis (TGA) enabled the examination of the thermal stability of prepared compositions. Table 2 presents that adding cellulose fibers to neat ENR made the composite more thermally stable than adding only PLA. ENR–CEL lost 5% of its weight at a temperature of 332 °C, while the temperature of 5% ENR–PLA weight loss was at 262 °C. The specimen ENR–CEL presented better thermal stability than compositions with antioxidants until 400 °C was reached. Above that temperature, the blend enriched with eugenol became the most thermally stable and lost 90% of its mass at 476 °C, while the ENR–CEL blend lost the same amount of mass at 473 °C. Comparing the overall impact of temperature, in the range of 0–350 °C, on composites containing selected antioxidants, material containing catechin hydrate seems to be the most resistant to high temperatures. ENR–CAT lost 15% of its mass at 348 °C. In the case of blends containing eugenol and flavone, a 15% loss of mass occurred at 342 °C and 340 °C, respectively. Moreover, compositions with natural additives presented thermal stability similar to the values presented by the ENR–PLA specimen.

Overall, the ENR–CEL was the most thermally stable among all specimens. Its thermal decomposition occurred in three steps. In opposition to this material, other samples have degraded in four steps. The T_2%_ and T_5%_ values of studied specimens differed up to 70 °C and 69 °C, respectively, which suggests their different thermal stability. Concerning only materials with natural additives, the ENR–CAT sample presented the finest stability, similar to the case of mechanical properties. Possibly, catechin hydrate cross-linked with cellulose particles, so it was more resistant to thermal degradation. On the other hand, the ENR–FLA material presented the worst thermal properties. It correlates with its relatively high value of -color-change parameter after aging.

### 2.4. FT-IR Absorbance Spectra Analysis

FT-IR spectra were examined in order to study the chemical structure of the composites’ surfaces and analyze changes that appeared during the solar-aging procedure (Figure 5 and Figure 6). Several significant changes in absorbance signals could be detected. Firstly, there was a change in peak intensity at ~3400 cm^−1^ (O-H group) in the case of every specimen. This signal was more intense after an aging process, which can be explained by the presence of adsorbed water, the formation of unstable hydroperoxides and alcohols, as a result of the oxidation process of the materials [42]. Subsequently, there were changes in ~2900 cm^−1^ (C-H stretching—groups present in ENR-50 [43]). Aging processes caused a significant decrease in the intensity of these peaks, which represented C-H moieties. ENR–CAT composition was an exception because changes in C-H groups were relatively low. It suggests suitable resistance to solar aging of this composite. Surface free energy calculations correspond with these conclusions, because total surface free energy of ENR–CAT did not present change as a result of solar aging. At ~1750 cm^−1^, very representative peaks assigned to stretching vibrations of C=O carbonyl groups were noted. Due to the solar-aging process, the absorbance of this peak increased in the case of every specimen. It results from the fact that the mechanism of polymer aging consists of the oxidation of the polymer chain and the creation of carbonyl groups [44]. However, compositions with catechin hydrate, eugenol and flavone presented minor differences in the intensity of this peak, so it suggests that these substances slowed down the aging process, because natural additives presented their antioxidative activity. Peaks at ~1450 cm^−1^ and ~1380 cm^−1^ were identified as -CH_3_ groups [45,46]. During the solar aging, the absorbance of these peaks decreased in all samples with the exception of the ENR–CAT blend. This indicates that catechin hydrate had a stabilizing effect on the surface of ENR-based material. Maximum visible at ~1080 cm^−1^ was related to ester bonds (C-O-C, C-O) characteristic for, e.g., PLA [47]. As there is a possibility of the formation of new ester bonds during the aging process, the absorbance of these bands increased significantly after aging in every specimen except for ENR–FLA, which presented nearly the same value of absorbance. The last significant band, located at ~890–860 cm^−1^, was corresponding to oxirane groups present in ENR-50. Polymers’ aging can be also indicated by the higher intensity of this peak. It should be highlighted that this peak was more intense in ENR–PLA and stayed almost unchanged in the case of ENR–CAT, which confirms once again the antioxidative activity of the catechin hydrate that stabilized the material.

Based on infrared spectroscopy, the carbonyl index (CI) was calculated for each specimen to analyze more precisely changes that occurred during solar aging and conclude the extent of polymer degradation (Figure 7). The degradation process may result in the cracking of C–H bonds and accompanying formation of hydroxyl and carbonyl groups. The value of CI for ENR–CEL before aging was close to zero, because in this sample any chemical compound included in its structure carbonyl group C=O. In other samples, the initial value of CI was a consequence of carbonyl moieties present in PLA and phytochemicals’ structures. It can be observed that CI increased for all samples during aging. It is a result of occurrent degradation which mechanism consists in creating new carbonyl groups.

The least rise was calculated for the ENR–CAT specimen (from 0.80 to 1.56). It indicates the lowest degree of degradation of this sample among other studied specimens and, as in the case of mechanical properties and color-change measurements, confirms the suitable level of catechin’s antioxidative activity.

On the other hand, the ENR–PLA material presented the biggest rise in CI (from 0.80 to 4.20). The reason for that is the absence of any antioxidant in this sample that could limit the advance of occurring degradation. The change of its CI is connected with the deterioration of mechanical properties, which was aforementioned in Section 2.2.

The carbonyl index of the ENR–FLA specimen rose from 0.60 to 4.00. In conjunction with the change of color parameter equal to 14.83 a.u., it also suggests the oxidation of polymer’s surface that occurred as a result of solar aging.

### 2.5. Surface Free Energy (SFE) Measurements and Optical Microscopy

To analyze the impact of solar aging on the samples’ surface properties, contact angle values for three measuring liquids (diiodomethane, ethylene glycol and distilled water) were used to calculate the surface free energy (SFE) of studied composites (Figure 8). Both dispersive and polar parts of SFE were taken into consideration. It can be noted that before aging, all polymer samples presented total SFE in the range of 20–30 mJ/m^2^. The polar part of SFE was lower than 5 mJ/m^2^ for all materials. Aging processes caused changes in SFE values—total SFE and its polar part increased. What is more, the biggest change in SFE values could be observed for samples without natural additives—it means that ENR–CEL and ENR–PLA degraded faster than other specimens.

The ENR–CAT sample presented distinct properties compared with other specimens. Its total SFE value remained almost unaltered after aging. It happened because adding catechin hydrate to ENR stabilized the material and protected the surface from degradation. In connection with FT-IR spectroscopy results described in this paper and elsewhere [48], there is the likelihood that particles of catechin hydrate migrated onto the surface of the polymer (Figure 9). Moreover, the phenomenon of the polymer’s surface oxidation and poor dissolution of the additive in the polymer is also probable. The increase in SFE value in other specimens can be explained with the micro-modifications of the specimens’ surfaces—scratches, cracks or pores [49]. Furthermore, the rise of the polar part of SFE after aging is a result of polymer oxidation, absorption of water (higher hydrophilicity) and the presence of more polar groups deriving from, e.g., added flavonoids. It should be also highlighted that in the case of the ENR + flavone composite, the polar part of SFE stayed almost unchanged after aging. It can suggest that this natural antioxidant effectively slowed down the aging process of the material by preventing it from creating oxidized chemical moieties.

To better understand changes concerning the surface properties, optical microscopy measurements were performed (Figure 10). As it can be seen, reference samples presented organized linear patterns on the surface. They were all brownish. After solar aging, there were visible significant changes recorded by the microscope. Surfaces of all ENR-based composites were characterized by abruptions, cracks and scratches. Most likely it was a reason for the SFE increase after aging. What is more, in samples with natural additives, pores or supposed air bubbles were visible. It suggests some inhomogeneity of studied materials.

### 2.6. Oxidation Induction Time (OIT)

With the aim of checking the thermal stabilization degree of prepared samples, the oxidation induction time (OIT) values were measured. Table 3 shows that adding eugenol and flavone to the basic mixtures did not improve thermal stability. Only the presence of catechin hydrate (OIT for ENR–CAT equal to 1.05 min.) delayed the onset of the thermal oxidation in comparison with the ENR–PLA sample, for which the onset OIT value was equal to 0.70 min. Catechin hydrate had a significant oxidative retardant effect on the polymer matrix compared to other chosen flavonoids because of having several hydroxyl groups in its chemical structure. Antioxidative activity of added catechin hydrate was especially apparent when the peak of OIT was studied; then the value for ENR–CAT was 12.50 min and it was about four times higher than the peak for ENR–CEL equal to 3.18 min.

In order to analyze the antioxidative activity of catechin hydrate, an experiment investigating the electrooxidation process of this flavonoid was presented by Masek et al. [50]. The described mechanism was considered an irreversible two-step process. Having analyzed the distribution of electron charges in the catechin, hydroxyl groups in ring B have been determined as the most susceptible to electrooxidation. In the presented mechanism of catechin electrooxidation, the exchange of two protons and one electron occurs at first and, as a result, semiquinone is formed. The next step includes the exchange of a second electron and a quinone is formed. The scheme describing the abovementioned mechanism is present in [50].

### 2.7. UV-Vis Spectra Analysis

The UV-Vis spectroscopy showed changes in composites’ structures after adding natural additives to basic compositions (Figure 11). Firstly, the absorbance characteristic for ENR-50 at 250 nm and 300 nm was visible [51]. Similarly, absorbance at 240 nm is a feature of HPMC and PLA [52,53]. The ENR–CAT UV-Vis spectra had a broad peak between 300 and 600 nm. According to Nagarajan et al. [54], a broad peak between 300 nm and 550 nm is typical for an oligomeric form of catechins. It suggests that polymerization of catechin hydrate occurred. In the case of the specimen with eugenol before aging, there was a specific inflection point on the graph at 280 nm—it is an absorbance characteristic for eugenol [55]. The above analysis of presented UV-Vis spectra confirms the presence of added phytochemicals in the structure of created mixtures. In regard to the impact of solar aging in every specimen with the exception of ENR–CAT and ENR–EUG, there was an increase in absorbance at around 280 nm. This wavelength is specific for ketones [56], so it may suggest the creation of new carbonyl groups as a result of aging processes. Consequently, figures presenting UV-Vis spectra of specimens with catechin hydrate and eugenol indicate antioxidative features of these compositions.

### 2.8. Antibacterial Experiment

Test strains of *Escherichia coli* were used in order to examine the antibacterial activity of the polymer’s surface (Figure 12). Surprisingly, the materials without antioxidants presented better antibacterial activity than materials with natural additives. It seems that phytochemicals lost their antibacterial activity during preparation of studied materials. What is more, the aging process made the antibacterial activity of samples worse than it was for unaged materials.

## 3. Materials and Methods

### 3.1. Materials and Processing

The studied composites were prepared using 50% epoxidized natural rubber (Muang Mai Guthrie Public Limited Company (Phuket, Thailand) (ENR-50)), polylactide (PLA) (IngeoTM Biopolymer 4043D) from Nature Works (Minnetonka, MN, USA) and powdered cellulose (ARBOCEL^®^ CE 2910 HE 50 LV, JRS GmbH, Rosenborg, Germany). Eugenol (98%), (+)-catechin hydrate (≥98% (HPLC), powder) and flavone were supplied by Sigma-Aldrich (Saint Louis, MO, USA). The curing agent-dicumyl peroxide (DCP, bis (α,α-dimethylbenzyl )peroxide), 98% of purity, was purchased from Merck (Darmstadt, Germany).

At first, the chosen substances, according to the proportions shown in Table 4, were mixed using a laboratory micromixer (Brabender Lab-Station from Plasti-Corder with the Julabo cooling system (Duisburg, Germany)) with a speed of 70 rpm for 30 mins and at the temperature of 180 °C. After that, samples were put between two roll mills (friction 1:1.1) for two mins. Finally, plate-like samples were formed in a hydraulic press at a temperature of 160 °C for 30 mins (pressure: 125 bar).

### 3.2. Color Identification

A Spectrophotometer UV-VIS CM-36001 (Konica Minolta Sensing, Inc., Osaka, Japan) based on the PN-EN ISO 105-J01 standard was used to measure the color of the studied specimens. To describe colors, the CIE-Lab system was used (a—red-green, b—yellow-blue, L—lightness). Therefore, color difference (ΔE), whiteness index (W_i_), chroma (C_ab_) and hue angle (h_ab_) values were counted for unaged and aged compositions according to Equations (1)–(4):(1)$$∆E=\sqrt{∆a^{2}+∆b^{2}+∆L^{2}}$$
(2)$$W_{i}=100\;-\;\sqrt{a^{2}{+b}^{2}{+(100\;-\;L)}^{2}}$$
(3)$$C_{\mathrm{ab}}=\sqrt{a^{2}{+b}^{2}}$$
(4)$$h_{\mathrm{ab}}\{\begin{array}{c} \mathrm{arctg}\left(\frac{b}{a}\right),\;\mathrm{when}\;a\;>\;0\;\wedge \;b\;>\;0\\ 180{}^{\circ}+\mathrm{arctg}\left(\frac{b}{a}\right),\;\mathrm{when}\;(a\;<\;0\;\wedge \;b\;>\;0)\;\vee \;(a\;<\;0\;\wedge \;b\;<\;0)\\ 360{}^{\circ}+\mathrm{arctg}\left(\frac{b}{a}\right),\;\mathrm{when}\;a\;>\;0\;\wedge \;b\;<\;0 \end{array}$$

### 3.3. Static Mechanical Tests

Elongation at break (Eb) and tensile strength (TS) were measured on the basis of the ISO 37 standards. A Zwick-Roell 1435 device (Ulm, Germany) was used to that end. Dumbbell-shaped specimens were formed to carry out tests (thickness—1 mm, total length—75 mm, width—12.5 mm). The speed of the samples’ stretch was 500 mm/min. Next, aging coefficients were calculated. The calculations were based on the Equations (5a)–(5c):(5a)$$K=\frac{{\left(\mathrm{TS}\times \mathrm{Eb}\right)}_{\mathrm{after}\;\mathrm{aging}}}{{\left(\mathrm{TS}\times \mathrm{Eb}\right)}_{\mathrm{before}\;\mathrm{aging}}}$$
(5b)$$K1=\frac{{\left(\mathrm{TS}\right)}_{\mathrm{after}\;\mathrm{aging}}}{{\left(\mathrm{TS}\right)}_{\mathrm{before}\;\mathrm{aging}}}$$
(5c)$$K2=\frac{{\left(\mathrm{Eb}\right)}_{\mathrm{after}\;\mathrm{aging}}}{{\left(\mathrm{Eb}\right)}_{\mathrm{before}\;\mathrm{aging}}}$$

### 3.4. Thermogravimetric Analysis (TGA)

The measurement was conducted with the use of a Mettler Toledo TGA/DSC 1 STARe System equipped with a Gas Controller GC10 (Greifensee, Switzerland). The test was performed with a heating rate of 20 °C/min in a temperature range of 25–1000 °C and air flow of 50 cm^3^/min. Studied specimens were located in alumina vessels.

### 3.5. Fourier-Transform Infrared Spectroscopy (FT-IR)

The measurements of Fourier-transform infrared spectroscopy (FT-IR) were performed with the use of a Thermo Fisher Scientific Nicolet 6700 FT-IR spectrometer supplied with a diamond Smart Orbit ATR sampling device (Waltham, MA, USA). The absorbance spectra were measured within the 4000–400 cm^−1^ range (absorption mode, 64 scans).

Carbonyl Index was calculated according to Equation (6):(6)$$\mathrm{CI}=\frac{{\mathrm{absorbance}}_{{C=O\;(\sim 1750\;\mathrm{cm}}^{-1})}}{{\mathrm{absorbance}}_{{C-H\;(\sim 2900\;\mathrm{cm}}^{-1})}}$$

### 3.6. Surface Free Energy (SFE) and Optical Microscopy

Calculations of surface free energy were based on contact angle measurements for ethylene glycol, 1,4-diiodomethane and distilled water (droplet of the volume approximately 1 μL). An OCA 15EC goniometer by DataPhysics Instruments GmbH (Filderstadt, Germany) supplied with dosing system (0.01–1 mL The Braun DS-D 1000 SF syringe) was used. SFE was calculated with the Owens–Wendt–Rabel–Kaelble (OWRK) method.

Optical microscopy measurements were performed with the use of an OPTA-TECH LAB-40 optical microscope and Capture V2.0 software. The specimens were observed with 50× magnification.

### 3.7. Oxidation Induction Time (OIT)

Oxidative induction time (OIT) parameter was obtained from a Mettler Toledo DSC (Greifensee, Switzerland) apparatus to evaluate an extent of thermal stabilization. Three OIT times were specified: onset, peak and endset. Isothermal condtions were kept. Samples with a weigth of 5–7 mg were heated from the room temperature to 220 °C under an air atmosphere. The measurement lasted 100 min.

### 3.8. UV-Vis Spectroscopy

A UV-Vis spectrophotometer (Evolution 220, Thermo Fisher Scientific, Waltham, MA, USA) was used to record the spectra of samples at wavelengths of 190–1100 nm.

### 3.9. Solar Aging Process

An Atlas SC 340 MHG Solar Simulator climate chamber (AMETEK Inc., Berwyn, IL, USA) provided with a 2500 W MHG lamp was employed in order to conduct the solar-aging process. A unique range of solar radiation is given by a special rare-earth halogen lamp. The samples were aged for 200 h at a temperature of T = 70 °C with the radiation intensity equal to 1200 W/m^2^ at 100% lamp power intensity.

### 3.10. Antibacterial Experiment

The degree of *Escherichia coli* bacteria viability after having contact with the surface of studied materials was examined. Firstly, 20 μL of bacteria suspension was put on a surface of a sample. Then it was incubated at 37 °C for 60 min. After this time, the applied suspension was collected from the surface and subjected to the procedure of staining with the “Viability/Cytotoxicity Assay kit for Bacteria Live and Dead Cells” staining kit (Biotium, Fremont, CA, USA). After incubation with fluorescent dyes, the percentage of live and dead bacteria was measured using the flow cytometry technique. BD CSampler software was used to evaluate the obtained results. Two control trials were made—positive (dead cells of microorganisms subjected to the activity of 98% ethanol) and negative (unmodified suspension of microorganisms). Three measurements for every material were made, and this cycle was repeated three times.

## 4. Conclusions

The obtained results present the influence of natural additives on physico–chemical properties of the ENR-50-based composites under conditions of solar aging. Selected natural antioxidants (catechin hydrate, eugenol and flavone) were used as potential stabilizers for polymers.

It should be highlighted that among all samples, the ENR-based composite with flavone presented the highest color change after aging, visible to the human eye (dE = 14.83 a.u.). It indicates the possibility of using this natural additive as a -color-aging indicator in active packaging. On the other hand, the aging process had low impact on the ENR–CAT color parameters (dE < 3 a.u.), which suggests its suitable resistance to degradation.

Static mechanical tests revealed that as a result of solar-aging conditions, all studied samples presented worse mechanical properties than before aging. Selected antioxidants only prevented more significant changes in TS and Eb values. Moreover, because of the solar aging, the elongation at break values of all samples were more affected than tensile strength values. It is clearly shown by the results of the ENR–FLA sample that became stiff and lost much of its flexibility. The ENR–CAT was marked by the highest value of aging coefficient out of all specimens (K = 0.38 a.u.) and presented enhanced mechanical properties in comparison with the reference sample ENR–CEL. The presence of catechin hydrate considerably slowed down the advance of a composite’s degradation. Materials presented in this paper can be applied in solutions that require on a long-term basis stable tensile-strength parameters, but invariable elongation at break values are not essential.

It is noteworthy that the ENR–CAT was characterized by the lowest carbonyl index value after aging (CI = 1.56) and the highest value of oxidation induction time (peak, 12.50 min) out of studied materials. It could be caused by the fact that catechin hydrate includes five hydroxyl groups in its chemical structure that participate in the mechanism of oxidation.

Additionally, studied antioxidants did not improve the thermal properties of polymeric material—the ENR–CEL sample was the most thermally stable. What is more, the presence of natural additives lowered antibacterial activity of studied materials in comparison with the ENR–CEL and ENR–PLA reference samples.

Results presented in this paper provide valuable information concerning the antioxidative activity of selected natural additives. The analysis of their impact on the polymer matrix under conditions of solar aging broadens the studies concerning polymers. In particular, the material containing catechin hydrate was marked by desired properties in the context of aging processes. The antioxidative activity of catechin hydrate is worth being submitted to further investigation.

Finally, studied composites seem to be eco-friendly as they were made with natural substances. These additives have potential to be used in the packaging industry where there is a need for materials that can stabilize polymer materials or act as aging indicators.

## Figures and Tables

**Figure 1 molecules-27-07877-f001:**
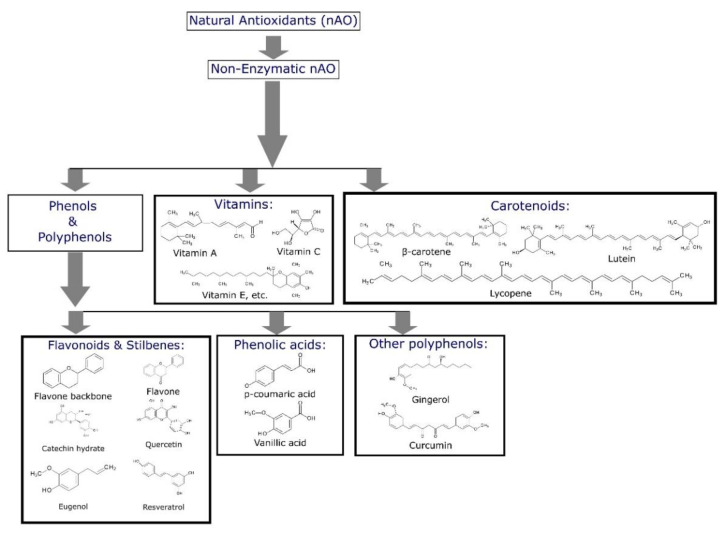
Classification and basic chemical structures of antioxidants applied in polymers [20].

**Figure 2 molecules-27-07877-f002:**
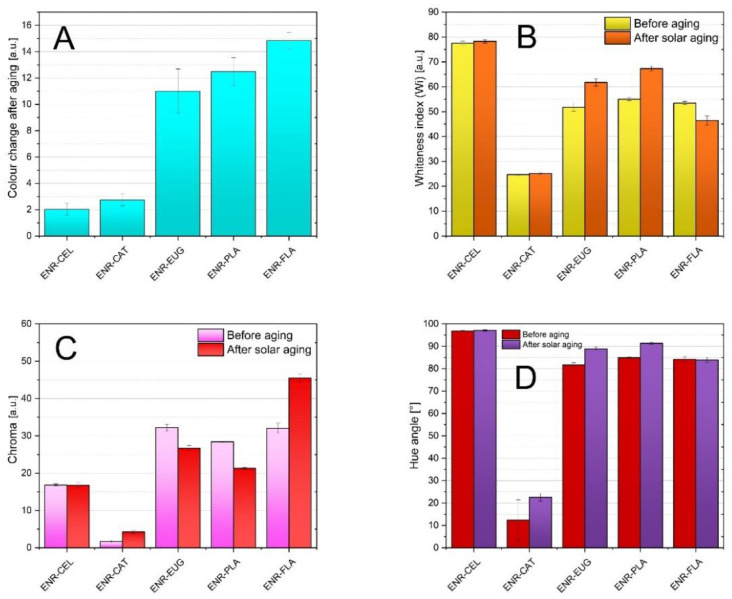
The color change (**A**), whiteness index (**B**), chroma values (**C**) and hue angles (**D**) of ENR-based composites.

**Figure 3 molecules-27-07877-f003:**
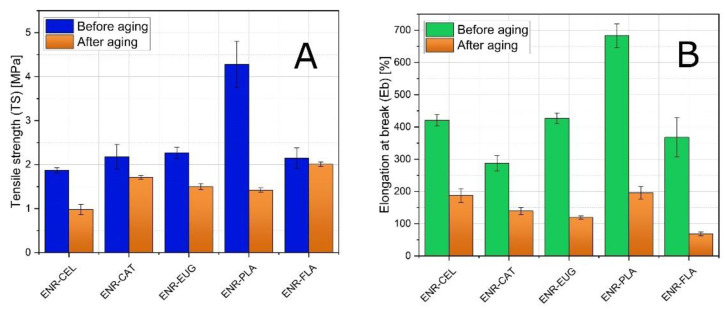
Tensile strength results (TS) (**A**), elongation at break (Eb) (**B**) of ENR-based mixtures with additional content.

**Figure 4 molecules-27-07877-f004:**
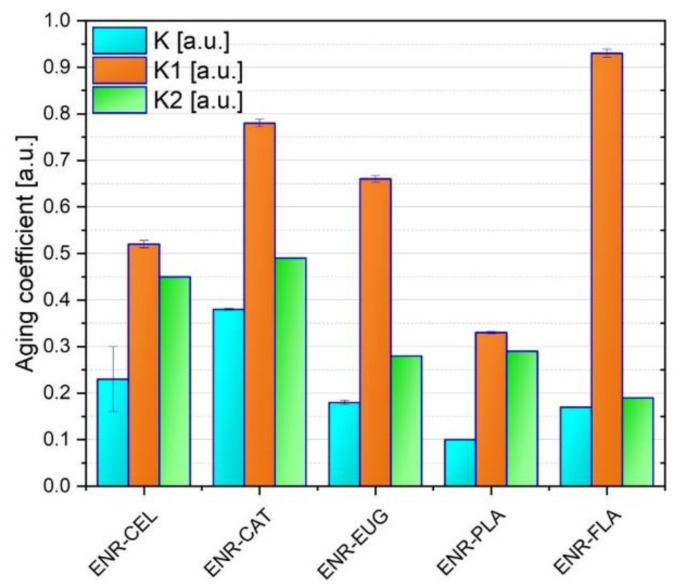
Aging coefficients of studied ENR-based compositions.

**Figure 5 molecules-27-07877-f005:**
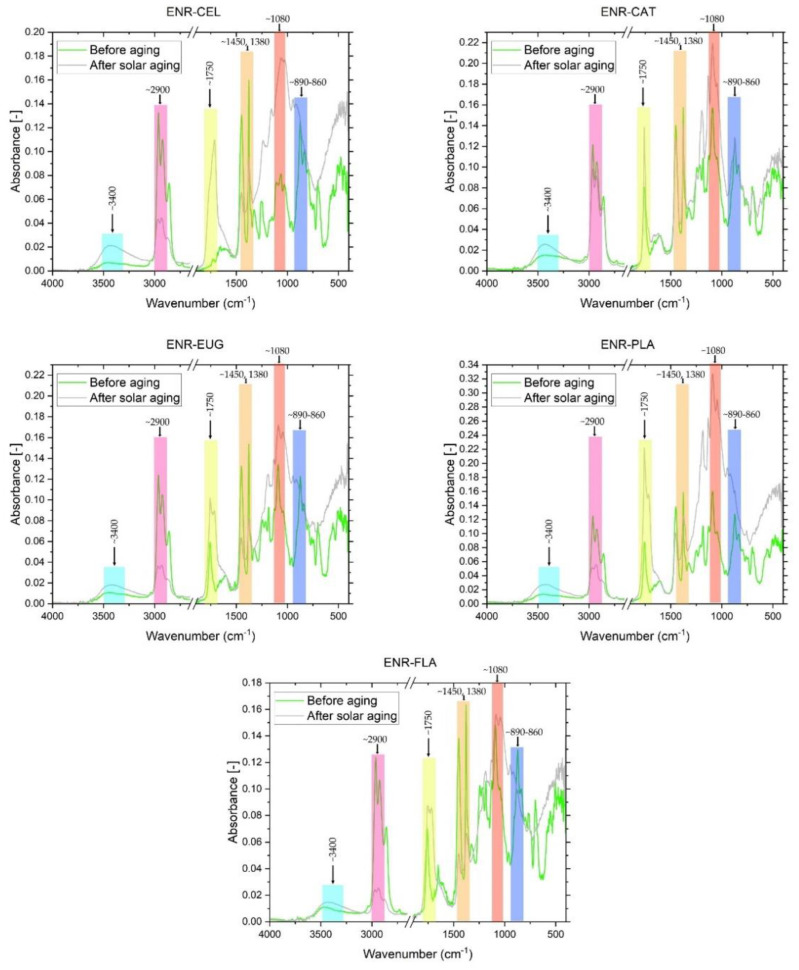
FT-IR spectra of ENR-50 blends with antioxidants before and after solar aging.

**Figure 6 molecules-27-07877-f006:**
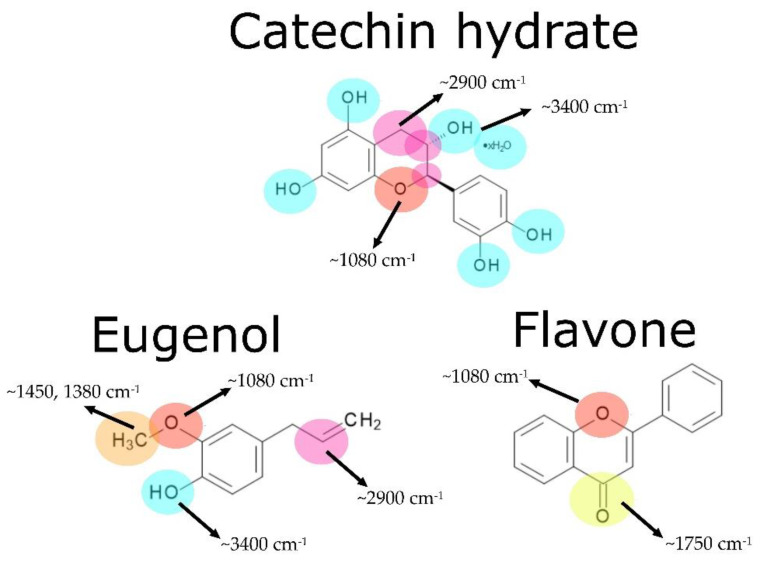
Absorption bands assigned to the chemical groups (bonds) that occur in the natural antioxidants: catechin hydrate, eugenol and flavone.

**Figure 7 molecules-27-07877-f007:**
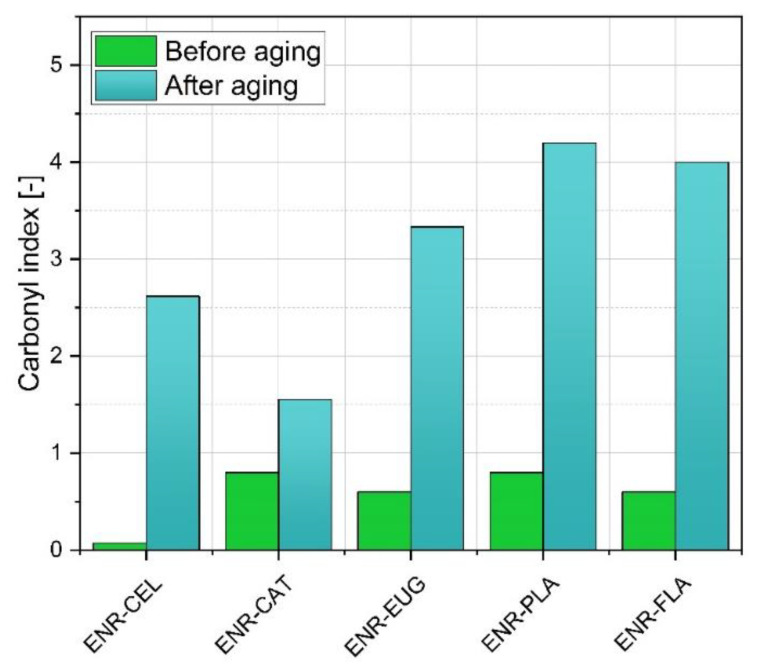
Carbonyl index (CI) values obtained for ENR-based composites before and after solar aging, where C=O groups were observed within the 1750 cm^−1^ range and CH groups at 2900 cm^−1^, respectively.

**Figure 8 molecules-27-07877-f008:**
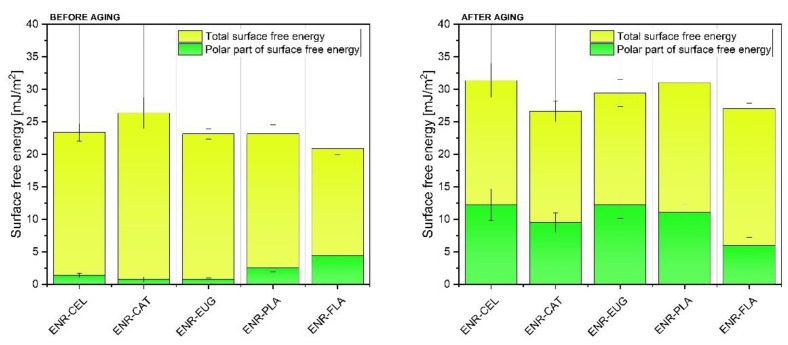
Surface free energy of ENR samples before and after aging.

**Figure 9 molecules-27-07877-f009:**
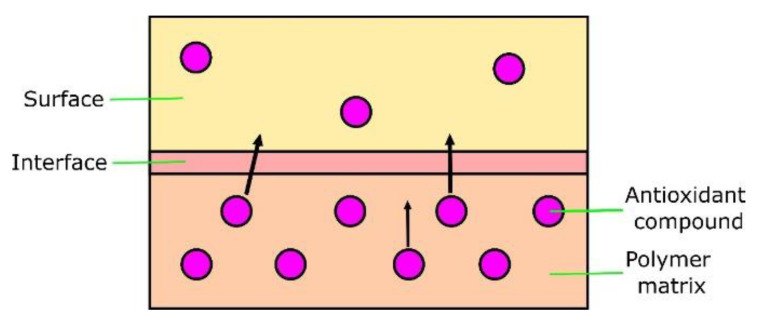
Scheme presenting possible migration of antioxidants onto the polymer surface.

**Figure 10 molecules-27-07877-f010:**
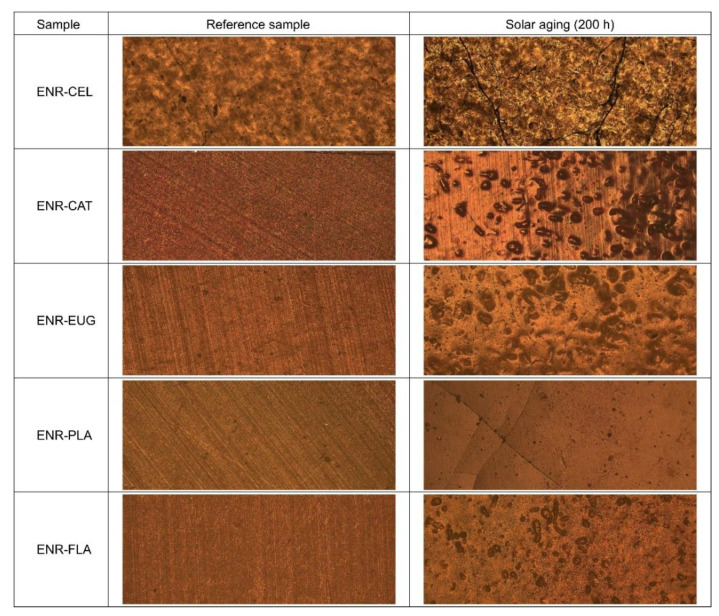
Effect of solar aging on the surface properties of ENR-based composites examined by optical microscopy at 50× magnification.

**Figure 11 molecules-27-07877-f011:**
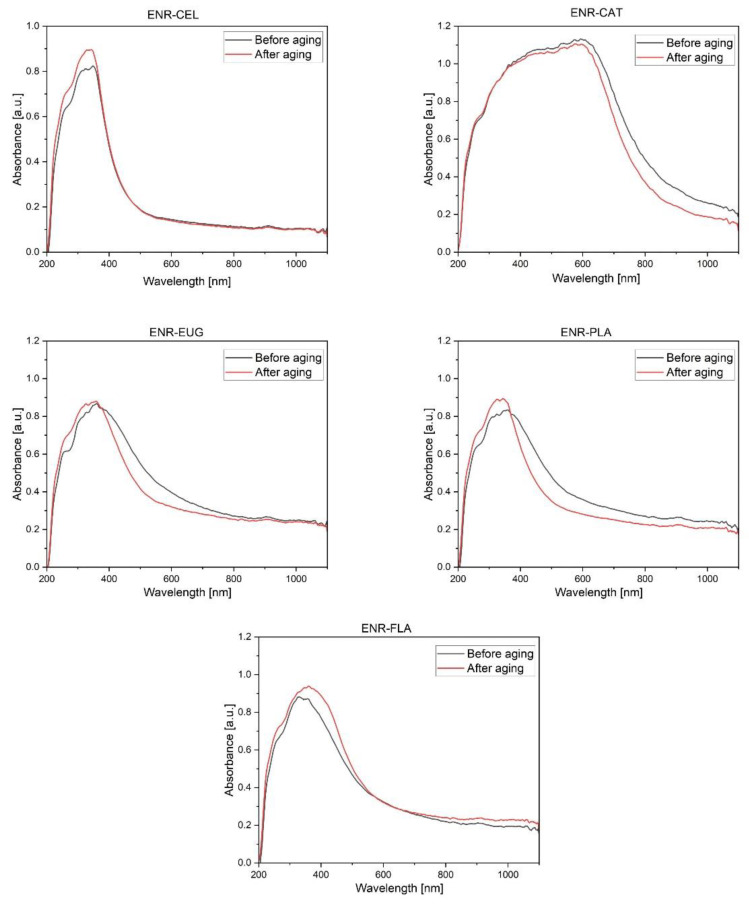
UV-Vis spectra recorded in the range of 200–1100 nm obtained for the studied ENR-based composites with extra content (HPMC, PLA, antioxidants).

**Figure 12 molecules-27-07877-f012:**
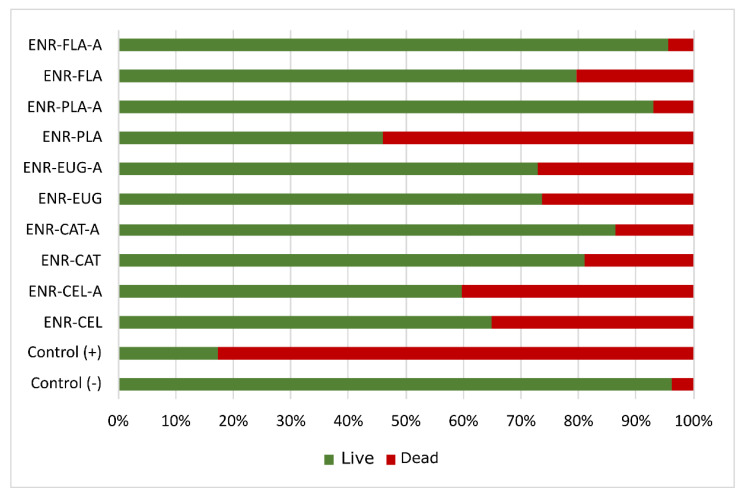
Results of antimicrobial tests performed for studied ENR-based composite samples. Letter “A” next to the sample symbol means “after aging”. “Dead” and “Live” refer to bacteria.

**Table 1 molecules-27-07877-t001:** Aging coefficients (K) of studied ENR-based compositions.

Sample	ENR–CEL	ENR–CAT	ENR–EUG	ENR–PLA	ENR–FLA
K	0.23	0.38	0.18	0.10	0.17

**Table 2 molecules-27-07877-t002:** Temperatures of the tested 50% epoxidized natural rubber (ENR-50) samples weight loss. T_x%_ is the temperature at which the weight loss is x%.

Sample	T_2%_ (°C)	T_5%_ (°C)	T_10%_ (°C)	T_20%_ (°C)	T_50%_ (°C)	T_90%_ (°C)
ENR–CEL	297	332	358	377	401	473
ENR–CAT	244	284	329	356	396	468
ENR–EUG	228	262	318	356	393	476
ENR–PLA	233	262	321	361	404	460
ENR–FLA	228	262	310	353	396	460

**Table 3 molecules-27-07877-t003:** Oxidation induction time values of ENR-based samples before aging and their energy of oxidation.

Sample	Onset [min]	Peak [min]	Endset [min]	Energy of Oxidation [J/g]
ENR–CEL	1.16	3.18	20.15	108
ENR–CAT	1.05	12.50	16.69	198
ENR–EUG	0.43	1.73	23.73	337
ENR–PLA	0.70	2.72	22.15	247
ENR–FLA	0.56	2.17	6.58	289

**Table 4 molecules-27-07877-t004:** Composition of the polymer blend mixtures prepared for analysis in this research. Abbreviations: ENR—epoxidized natural rubber, PLA—poly(lactic acid), CEL—cellulose, CAT—catechin hydrate, EUG—eugenol, FLA—flavone, DCP- bis (α,α-dimethylbenzyl)peroxide, phr—per hundred rubber (which means—there are “x” parts by weight of the substance for one hundred parts by weight of rubber).

Sample	Polymer Mixture Composition [phr]						
	ENR	CEL	PLA	CAT	EUG	FLA	DCP
ENR–CEL	100	15	-	-	-	-	1.5
ENR–CAT	100	15	10	2	-	-	1.5
ENR–EUG	100	15	10	-	2	-	1.5
ENR–PLA	100	-	10	-	-	-	1.5
ENR–FLA	100	15	10	-	-	2	1.5

## Data Availability

Not applicable.

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
