# Peer review of "Biocomposites of Epoxidized Natural Rubber Modified with Natural Substances"

_molecules, 2022, doi:10.3390/molecules27227877_

Round 1
Reviewer 1 Report
The article is well written and it is within the scope of the journal but I’m not convinced that it has enough originality. Natural antioxidants used in this paper have already been presented in many articles. It is known that eugenol use as an efficient thermal stabilizer.
I don't understend why the author gives the results of another works (co-author) in the results (lines 204, 383). Was it just a comparison?
I don't agree that have potential to be used in packaging industry becouse eugenol has very intensive smell (even after materials processing). What happens to this smell after the material has aged?
This paper describes ENR/PLA blends with the addition catechin hydrate, eugenol and flavone after solar aging. Only analysis of their impact on the polymer matrix under conditions of solar aging brings something new.
Author Response
Institute of Polymer and Dye Technology
Technical University of Lodz
90-924 Lodz, ul Stefanowskiego 12/16, Poland
Tel.: +48 42 631 32 23, Fax: +48 42 636 25 43
November 1, 2021
Molecules (MDPI)
Dear Editors,
We are resubmitting our revised paper entitled Biocomposites of Epoxidized Natural Rubber Modified with Natural Substances by Konrad Stefaniak and Anna Masek with a request to reconsider it for publication in Molecules (MDPI). We have carefully considered the Reviewers comments. The manuscript was revised according to these comments. The list of responses to the reviewers comments and corrections made in the manuscript is attached.
The manuscript has not been previously published, is not currently submitted for review to any other journal and will not be submitted elsewhere before a decision is made by this journal.
For correspondence, please use the following information:
corresponding author: Anna Masek
Institute of Polymer and Dye Technology
Technical University of Lodz
90-924 Lodz, ul Stefanowskiego 12/16, Poland
Tel.: +48 42 631 32 93
Fax: +48 42 636 25 43
e-mail: anna.masek@p.lodz.pl
Yours sincerely,
Ph. D., D.Sc. Anna Masek
Responses to the Reviewer's 1 comments
Main corrections in the paper are marked by green colour through the whole paper.
Reviewer 1: The article is well written and it is within the scope of the journal but I’m not convinced that it has enough originality. Natural antioxidants used in this paper have already been presented in many articles. It is known that eugenol use as an efficient thermal stabilizer.
Answer: Thank you for your positive evaluation of our manuscript. The originality of presented article consists in the composition of obtained materials and analyzing the impact of added antioxidants on the polymer matrix under solar aging conditions. The novelty of this paper has been emphasized as follows:
“Compositions of created materials and studying the impact of additives on retarding the degradation processes of polymer matrix under conditions of solar aging are scientific novelties of this research”
Reviewer 1: I don't understand why the author gives the results of another works (co-author) in the results (lines 204, 383). Was it just a comparison?
Answer: We thank Reviewer for paying attention to this issue. The first reference has been removed. The second reference is presented in order to introduce and describe the mechanism of catechin electrooxidation process. The scheme describing this mechanism has been removed. The fragment of the text has been improved as follows:
“In order to analyse antioxidative activity of catechin hydrate an experiment investigating the electrooxidation process of this flavonoid was presented by Masek et al. [51]. Described mechanism was considered as an irreversible two-step process. Having analysed the distribution of electron charges in the catechin, hydroxyl groups in ring B have been determined as the most susceptible to electrooxidation. In presented mechanism of catechin electrooxidation, the exchange of two protons and one electron occurs at first and, as a result, semiquinone is formed. The next step includes the exchange of a second electron and a quinone is formed. Scheme describing abovementioned mechanism is present in reference [51].”
Reviewer 1: I don't agree that have potential to be used in packaging industry because eugenol has very intensive smell (even after materials processing). What happens to this smell after the material has aged?
Answer: We are thankful for this comment. In the future, we might analyze the topic of antioxidants’ smell by the use of ion mobility spectrometry (IMS).
After organoleptic analysis it was found that the intensive smell of eugenol was not present after material’s processing and aging. A following fragment of text has been added:
“Additionally, it should be highlighted that ENR-EUG sample has potential to be used in packaging industry despite a very intensive smell of pure eugenol. This smell is significantly reduced as a result of material processing. Eventually, it was verified via organoleptic analysis that aged samples have lost its smell and there is not any contraindication to usage in packaging industry.”
Reviewer 1: This paper describes ENR/PLA blends with the addition catechin hydrate, eugenol and flavone after solar aging. Only analysis of their impact on the polymer matrix under conditions of solar aging brings something new.
Answer: We are thankful for this comment. Compositions of presented ENR/PLA blends with natural additives and the analysis of antioxidants’ impact on the polymer matrix under conditions of solar aging are the novelties of this paper. It has been emphasized as follows:
“The obtained results present the influence of natural additives on physico-chemical properties of the ENR-50-based composites under conditions of solar aging. Selected natural antioxidants (catechin hydrate, eugenol, and flavone) were used as potential stabilizers for polymers.”
Reviewer 2 Report
This work prepared Biocomposites derived from polylactic acid with epoxidized natural rubber, and with the addition of three different phytochemicals. Overall, this paper is well-prepared with a good-quality image. However, several things should be addressed.
1. Introduction part needs to be broadened. Please consider discussing the green composite materials of a different compound such as PVA, gelatin, etc. Then explain why PLA is more likable in this paper.
2. Literature on the phytochemicals needs to be improved (add the number such as the improvement rate of the antimicrobial, tensile strength, etc.)
3. What is the motivation for aging treatment in the tensile test if the results of the "after aging" sample are decreased?
4. Conclusion needs to be briefly improved (method, the best sample, the best results). Please discuss the tensile test section in the conclusion and re-explain its correlation to the application.
5. Please consider citing this most recent Biocomposites paper. Two of them discussed Biocomposites with Uncaria Gambir (a traditional leaf that has catechin compound for up to 85%), and also has potential for packaging application:
https://www.mdpi.com/2504-477X/6/10/316
https://www.sciencedirect.com/science/article/pii/S014181302200410X
https://www.sciencedirect.com/science/article/pii/S014181302200410X
Author Response
Institute of Polymer and Dye Technology
Technical University of Lodz
90-924 Lodz, ul Stefanowskiego 12/16, Poland
Tel.: +48 42 631 32 23, Fax: +48 42 636 25 43
November 1, 2021
Molecules (MDPI)
Dear Editors,
We are resubmitting our revised paper entitled Biocomposites of Epoxidized Natural Rubber Modified with Natural Substances by Konrad Stefaniak and Anna Masek with a request to reconsider it for publication in Molecules (MDPI). We have carefully considered the Reviewers comments. The manuscript was revised according to these comments. The list of responses to the reviewers comments and corrections made in the manuscript is attached.
The manuscript has not been previously published, is not currently submitted for review to any other journal and will not be submitted elsewhere before a decision is made by this journal.
For correspondence, please use the following information:
corresponding author: Anna Masek
Institute of Polymer and Dye Technology
Technical University of Lodz
90-924 Lodz, ul Stefanowskiego 12/16, Poland
Tel.: +48 42 631 32 93
Fax: +48 42 636 25 43
e-mail: anna.masek@p.lodz.pl
Yours sincerely,
Ph. D., D.Sc. Anna Masek
Responses to the Reviewer's 2 comments
Main corrections in the paper are marked by green colour through the whole paper.
Reviewer 2: This work prepared Biocomposites derived from polylactic acid with epoxidized natural rubber, and with the addition of three different phytochemicals. Overall, this paper is well-prepared with a good-quality image. However, several things should be addressed.
Answer: Thank you for your positive evaluation of our manuscript and for all valuable comments that will improve the quality of the article.
Reviewer 2: 1. Introduction part needs to be broadened. Please consider discussing the green composite materials of a different compound such as PVA, gelatin, etc. Then explain why PLA is more likable in this paper.
Answer: We are grateful for this comment. The discussion and explanation have been added:
“Epoxidized natural rubber/poly(lactic acid) (ENR/PLA) blends are reported to be one of the most interesting composite materials [1–4]. Obviously there are many other different green composite materials based on compounds such as polyvinyl alcohol (PVA) [5], polycaprolactone (PCL) [6] and gelatin [7,8], but choosing PLA is preferable. It results from the derivation of each material. Both PVA and PCL are based on petroleum industry which is harmful to the environment [9,10]. Abundant gelatin sources are derived from animals [8], which could be used in food industry rather than polymer technology. However, PLA is a biodegradable and biocompatible aliphatic polyester which derives from renewable, plant-based resources and in that it stands out from other polymers as a bioplastic [11].”
Reviewer 2: 2. Literature on the phytochemicals needs to be improved (add the number such as the improvement rate of the antimicrobial, tensile strength, etc.)
Answer: We are thankful for drawing our attention to this issue. The literature on the phytochemicals has been improved as follows:
“After 30 h of irradiation Mn of neat PLA was lowered by circa 60 %, while PLA/resveratrol specimens retained approximately 75 % of their original molar mass. Furthermore, after about 60 h of irradiation, neat PLA specimens were unfit to be tested in terms of mechanical properties (they were too brittle to be handled) whereas PLA/resveratrol samples retained approximately 30 % of their initial ε.”
“Other interesting research concerned using α-tocopherol (vitamin E) as an adequate processing stabilizer against the thermo-oxidative degradation that occurs e.g. during the processing of poly(lactic acid) for food packaging [23,28,29]. It reported that adding α-tocopherol to polylactic acid/polyethylene glycol (PLA/PEG) films resulted in a 23.7 % decrease in water vapor permeability and 9.4 % increase in oxygen permeability in relation to PLA/PEG without α-tocopherol. This second observation limits using PLA/α-tocopherol films as efficient food packaging protectors. In addition, PLA/α-tocopherol films showed enhanced antibacterial and antioxidant features. Campoccia et al. [30] studied bacterial adhesion of PLA/α-tocopherol films. Results revealed that the incorporation of α-tocopherol in PLA films led to a 41 % decrease of S. aureus adhesion in Tryptose Broth medium after 24 h in comparison to neat PLA.”
“Adding eugenol to PVC resulted in a threefold increase in thermal stability compared to PVC stabilized by tin mercaptide (reference stabilizer).”
“Other studies were presented by Abral et al. [35]. Ultrasonicated cellulose-based biocomposites with an extract from Uncaria gambir (G) leaves, which contain polyphenols, predominantly catechin were obtained. Presented materials have potential for packaging application. The results showed that adding 0.05 g of G per gram of bacterial cellulose (BC) powder increased the value of specimen’s tensile strength from 72.9 MPa to 105.6 MPa. Moreover, adding 0.2 g of G per 1 g of BC powder enhanced the antimicrobial activity of the material (zone of inhibition against Escherichia coli raised from 0 mm to 6.8 mm).”
Reviewer 2: 3. What is the motivation for aging treatment in the tensile test if the results of the "after aging" sample are decreased?
Answer: We thank Reviewer for paying attention to this problem. The explanation has been given as follows:
“The aim of these tests was to study the impact of solar aging on specimens’ properties. Regardless of the findings, the obtained results give valuable information concerning the stabilizing activity of selected antioxidants.”
Reviewer 2: 4. Conclusion needs to be briefly improved (method, the best sample, the best results). Please discuss the tensile test section in the conclusion and re-explain its correlation to the application.
Answer: We are grateful for this comment. This section has been rewritten as follows:
“4. Conclusions
The obtained results present the influence of natural additives on physico-chemical properties of the ENR-50-based composites under conditions of solar aging. Selected natural antioxidants (catechin hydrate, eugenol, and flavone) were used as potential stabilizers for polymers.
It should be highlighted that among all samples ENR-based composite with flavone presented the highest colour change after aging, visible to the human eye (dE = 14.83 a.u.). It indicates the possibility of using this natural additive as a colour aging indicator in active packaging. On the other hand the aging process had low impact on the ENR-CAT colour parameters (dE < 3 a.u.) which suggests its suitable resistance to degradation.
Static mechanical tests revealed that as a result of solar aging conditions all studied samples presented worse mechanical properties than before aging. Selected antioxidants only prevented more significant changes in TS and Eb values. Moreover, because of the solar aging elongation at break values of all samples were more affected than tensile strength values. It is clearly showed by the results of ENR-FLA sample which became stiff and lost much of its flexibility. ENR-CAT was marked by the highest value of aging coefficient out of all specimens (K = 0.38 a.u.) and presented enhanced mechanical properties in compare with the reference sample ENR-CEL. The presence of catechin hydrate considerably slowed down the advance of composite’s degradation. Materials presented in this paper can be applied in solutions, which require on a long-term basis stable tensile strength parameter, but invariable elongation at break values are not essential.
Noteworthy, ENR-CAT was characterized by the lowest carbonyl index value after aging (CI = 1.56) and the highest value of oxidation induction time (peak, 12.50 min.) out of studied materials. It could be caused by the fact that catechin hydrate includes five hydroxyl groups in its chemical structure which participate in the mechanism of oxidation.
Additionally, studied antioxidants did not improve thermal properties of polymeric material – ENR-CEL sample was the most thermally stable. What is more, the presence of natural additives lowered antibacterial activity of studied materials in compare with ENR-CEL and ENR-PLA reference samples.
Results presented in this paper give a valuable information concerning the antioxidative activity of selected natural additives. The analysis of their impact on the polymer matrix under conditions of solar aging broadens the studies concerning polymers. Especially the material containing catechin hydrate is marked by desired properties in the context of aging processes. The antioxidative activity of catechin hydrate is worth to be submitted to a further investigation.
Finally, studied composites seem to be eco-friendly as they were made with natural substances. These additives have potential to be used in packaging industry where there is a need for materials which can stabilize polymer materials or act as aging indicators.”
Reviewer 2:
- Please consider citing this most recent Biocomposites paper. Two of them discussed Biocomposites with Uncaria Gambir (a traditional leaf that has catechin compound for up to 85%), and also has potential for packaging application:
https://www.mdpi.com/2504-477X/6/10/316
https://www.sciencedirect.com/science/article/pii/S014181302200410X
https://www.sciencedirect.com/science/article/pii/S014181302200410X
Answer: We are grateful for this valuable comment which may enrich our article. Proposed papers have been added to the references.
- Rahmadiawan, D.; Abral, H.; Yesa, W.H.; Handayani, D.; Sandrawati, N.; Sugiarti, E.; Muslimin, A.N.; Sapuan, S.M.; Ilyas, R.A. White Ginger Nanocellulose as Effective Reinforcement and Antimicrobial Polyvinyl Alcohol/ZnO Hybrid Biocomposite Films Additive for Food Packaging Applications. J. Compos. Sci. 2022, 6, 316, doi:10.3390/JCS6100316.
- Abral, H.; Kurniawan, A.; Rahmadiawan, D.; Handayani, D.; Sugiarti, E.; Muslimin, A.N. Highly antimicrobial and strong cellulose-based biocomposite film prepared with bacterial cellulose powders, Uncaria gambir, and ultrasonication treatment. Int. J. Biol. Macromol. 2022, 208, 88–96, doi:10.1016/J.IJBIOMAC.2022.02.154.
“Obviously there are many other different green composite materials based on compounds such as polyvinyl alcohol (PVA) [5], polycaprolactone (PCL) [6] and gelatin [7,8], but choosing PLA is preferable.”
“Other studies were presented by Abral et al. [35]. Ultrasonicated cellulose-based biocomposites with an extract from Uncaria gambir (G) leaves, which contain polyphenols, predominantly catechin were obtained. Presented materials have potential for packaging application. The results showed that adding 0.05 g of G per gram of bacterial cellulose (BC) powder increased the value of specimen’s tensile strength from 72.9 MPa to 105.6 MPa. Moreover, adding 0.2 g of G per 1 g of BC powder enhanced the antimicrobial activity of the material (zone of inhibition against Escherichia coli raised from 0 mm to 6.8 mm).”